# Sperm and Oocyte Chromosomal Abnormalities

**DOI:** 10.3390/biom13061010

**Published:** 2023-06-17

**Authors:** Osamu Samura, Yoshiharu Nakaoka, Norio Miharu

**Affiliations:** 1Department of Obstetrics and Gynecology, The Jikei University School of Medicine, Tokyo 105-8461, Japan; 2Department of Obstetrics and Gynecology, IVF Namba Clinic, Osaka 550-0015, Japan; 3Department of Clinical Genetics, Hiroshima Red Cross Hospital & Atomic-bomb Survivors Hospital, Hiroshima 730-0052, Japan

**Keywords:** sperm chromosome, aneuploidy, oocyte, gametogenesis, meiosis

## Abstract

Gametogenesis, the process of producing gametes, differs significantly between oocytes and sperm. Most oocytes have chromosomal aneuploidies, indicating that chromosomal aberrations in miscarried and newborn infants are of oocyte origin. Conversely, most structural anomalies are of sperm origin. A prolonged meiotic period caused by increasing female age is responsible for an increased number of chromosomal aberrations. Sperm chromosomes are difficult to analyze because they cannot be evaluated using somatic cell chromosome analysis methods. Nevertheless, researchers have developed methods for chromosome analysis of sperm using the fluorescence in situ hybridization method, hamster eggs, and mouse eggs, allowing for the cytogenetic evaluation of individual sperm. Reproductive medicine has allowed men with severe spermatogenic defects or chromosomal abnormalities to have children. However, using these techniques to achieve successful pregnancies results in higher rates of miscarriages and embryos with chromosomal abnormalities. This raises questions regarding which cases should undergo sperm chromosome analysis and how the results should be interpreted. Here, we reviewed clinical trials that have been reported on oocyte and sperm chromosome analyses. Examination of chromosomal abnormalities in gametes is critical in assisted reproductive technology. Therefore, it is necessary to continue to study the mechanism underlying gametic chromosomal abnormalities.

## 1. Introduction

Most chromosomal aberrations are known to occur during gametogenesis, especially in meiosis. Gametogenesis begins in the primordial germ cell, which undergoes repeated somatic cell divisions, and finally enters meiosis with replicated DNA. The early phase of meiosis I is called the prophase, which is divided into five stages: leptonema, zygonema, pachynema, diplonema, and diakinesis. During this process, chromosomes undergo a homology search, and homologous chromosomes pair to form the synaptonemal complex. Furthermore, recombination between homologous chromosomes to repair the DNA double-strand breaks (DSBs) on each chromosome caused by Spo11 or other proteins results in the formation of chiasmata [1]. These are important structures to create genetic diversity and subsequent accurate chromosome segregation. Most aneuploidies are of oocyte origin, as indicated by the origin of chromosomal aberrations in miscarried and newborn infants [2]. In contrast, most structural abnormalities are of sperm origin. Here, we reviewed various clinical trials reporting on oocyte and sperm chromosome analyses.

## 2. Sperm

### 2.1. History of Sperm Chromosome Analysis

Sperm chromosomes cannot be analyzed in the same manner as blood lymphocyte chromosomes. This is because sperm are primarily terminally differentiated cells and do not undergo cell division, during which chromosomes are most visible. In 1978, Rudak et al. [3] published the first sperm chromosome analysis method that took advantage of the penetration of human sperm into zona-free hamster eggs. However, the analysis faced problems with the success rate and quality of the obtained metaphase plate. Kamiguchi et al. made various improvements in Japan and established this as a revised method in 1986 [4]. Since this method uses only swim-up motile sperm for analysis, concerns have been raised regarding the possibility that it may not reflect chromosomal aberrations in all sperm. Later, intracytoplasmic sperm injection (ICSI) became available for chromosome analysis, enabling the random analysis of sperm in semen; however, only a few dozen sperm can be analyzed [5]. In the 1990s, a sperm chromosome analysis method using the fluorescence in situ hybridization (FISH) method was developed, making it possible to analyze aneuploidy in a large number of sperm. This method enables the analysis of multiple sperm simultaneously; however, its performance is affected by the interpretation and accuracy of the signals obtained from the DNA probe since the signals are analyzed visually. These methods are summarized in Table 1.

### 2.2. Results of Sperm Chromosome Analysis Using Zona-Free Hamster Eggs and Mouse Eggs 

According to a report by Kamiguchi et al. [4], utilizing the zona-free hamster egg method, 13.9% of normal males exhibit sperm chromosome anomalies, with aneuploidy accounting for 0.9% and structural aberrations accounting for 13%. Other reports [6,7] have shown that the rate of structural aberrations exceeds that of aneuploidy. The frequency of these abnormalities is higher than that in other animals. The development of the ICSI method for mouse eggs has enabled the analysis of not only motile sperm, but all sperm. Analysis of 477 normal human spermatozoa showed that the aneuploidy rate was 1.7%, whereas the structural abnormality rate was 8.8% [8]. These results are consistent with the hamster egg analysis mentioned earlier.

### 2.3. Sperm Chromosome Analysis Using Sperm-FISH

Since the 1990s, studies have been conducted on various sperm types using sperm-FISH. Unlike the conditions for conventional FISH study, which is performed on normal somatic cells, sperm nuclei are highly aggregated, making it difficult for probes to hybridize to sperm chromatin. Therefore, it is necessary to decondense sperm nuclei using chemicals. The results of sperm-FISH analysis differ among analysis centers because this is attributed to variations in the conditions used to determine and define normal spermatozoa. Automated analyzers are also being utilized to improve the accuracy of sperm-FISH analysis results [9].

### 2.4. Results of Chromosome Analysis in Normal Males

Previous studies [10] have primarily focused on chromosomes 13, 18, 21, X, Y, and a few autosomes. Notably, many investigators have performed analyses using probes specific for the 13, 18, 21, X and Y chromosomes because these trisomies for these three autosomes and aneuploidies of sex chromosomes are more viable than aneuploidies of other autosomes. In contrast, Saijuan Zhu [11] et al. examined all the chromosomes in normal spermatozoa from 10 donors and found that the average frequency of disomy sperm was 0.16 ± 0.08% for all chromosomes and 0.33% ± 0.16% for nullisomy sperm. They found no significant differences in the frequency of abnormalities among chromosomes.

### 2.5. Results of Chromosome Analysis in Men with Abnormal Semen Findings

The factors contributing to male infertility are low sperm count, low motility, and abnormal morphology. Consequently, numerous reports have documented the high rate of sperm chromosomal aberrations in men with infertility. However, all reports indicate that this rate of sperm chromosomal aberrations in infertile men is approximately three times higher than that in the general male population [12,13]. Abnormality rates are exceptionally high in cases of oligoasthenoteratozoospermia (OAT), where sperm count, motility, and morphology are poor, along with obstructive azoospermia, where sperm is collected from the testes [14,15]. The correlation between individual semen findings and abnormality rates has also been studied in various ways. 

Furthermore, sperm chromosomal abnormality rates tend to be higher with lower sperm concentration, sperm motility rate, and a higher rate of abnormal morphology [16,17]. The strongest correlation was observed with sperm concentration [18,19]. These results differ from those obtained using the ICSI of mouse oocytes, as described above. In the case of OAT, the number of sperm with disomy for chromosomes 13, 18, and 21 has been reported to be three times higher than that in fertile men, and the rate of XY disomy was eight times higher than that in fertile men. However, the number of spermatozoa in the testes did not differ significantly from that in normal sperm. Clinical prognosis after ICSI is poor in patients whose intracytoplasmic sperm is used because of a high rate of sperm chromosome aberrations [13].

### 2.6. Male Age and Sperm Chromosome Aberration Rate

As women age, it is widely recognized that the likelihood of aneuploidy in their eggs increases. However, the relationship between male age and sperm chromosome aberration rate remains inconclusive. Notably, several prior studies have shown no increase in the aneuploidy rate with male age [20,21], but a previous study has shown an increase in structural abnormalities [22]. In contrast, a few reports indicate a rise in aneuploidy rates [23,24,25]. In fact, most studies [23,24,25] have focused on chromosomes 21, 18, 13, X, and Y, where aneuploidy may exist. For instance, Donate et al. examined five healthy men under 40 and five healthy men over 60, using sperm-FISH with several p and q subtelomeric probes to analyze and compare chromosome aberration rates for a total of 19 chromosomes [26]. They discovered no significant differences in the frequencies of disomic and diploid sperm. Based on these studies, it remains unclear whether the rate of aneuploid chromosomal aberrations in sperm increases with age in men. Furthermore, most studies are limited by sample size or the throughput of the analysis. Further studies are needed to investigate this relationship between age and sperm chromosomal aberration rates in infertile men.

### 2.7. Sperm Chromosome Aberration Rates in Men with Chromosome Aberrations

#### 2.7.1. 47, XYY Males

XYY males have an extra Y chromosome due to sister chromatid non-disjunction in meiosis II. XYY males are typically fertile. In spermatozoa from males with sex chromosome aberrations, the investigation’s main focus has been numerical aberrations of the sex chromosomes [27]. A report summarizing previous research on the disomy of sex chromosomes (Table 2) revealed that XX disomy, YY disomy, and XY disomy were particularly frequent, with average frequencies of 1.65 ± 2.31%, 1.54 ± 1.53%, and 4.43 ± 6.03%, respectively. Nevertheless, mosaic 47, XYY/46, XY cases exhibited no difference in frequency of XX, XY, or YY disomy compared with males of normal karyotype [13]. 

Gonzalez et al. performed sperm chromosome analysis and preimplantation genetic diagnosis (PGD) of embryos obtained by ICSI from two 47, XYY men. They reported that the frequency of XY disomy sperm ranged from 16.7% to 19%. Moreover, they discovered that 32% of the 47 embryos had aneuploidy [28].

#### 2.7.2. 47, XXY Males

During meiosis I, there is an excess of the X chromosome in 47, XXY men, which is often lost during spermatogenesis. Approximately 3–11% of 47, XXY men are considered to be azoospermic; however, they may also experience oligozoospermia. Since these men often conceive using ICSI, the chromosomal composition of individual sperm becomes an issue. Notably, multiple studies on sperm chromosomes have been conducted (Table 2), with the frequency of XX disomy, YY disomy, and XY disomy averaging at 4.64% ± 2.56%, 0.30% ± 0.46%, and 11.1% ± 6.89%, respectively, further indicating a high frequency of XY disomy [13]. In addition, the frequency of XY disomy was higher than that of YY disomy. Chromosome 21 abnormalities have also been reported with diploidy and aneuploidy occurring at frequencies of 0.03% ± 4.2%, and 6.2%, respectively [22]. The frequency of XX disomy, YY disomy, and XY disomy in mosaic 47, XXY/46, XY cases was lower than these, averaging at 0.40% ± 0.31%, 0.42% ± 0.49%, and 1.22% ± 0.71%, respectively [13]. Published data to date show no chromosomal abnormalities in children born to Kleinfelter Syndrome (KS) patients, except in two cases [29]. The prevalence of chromosomal abnormalities in the general population is 0.5–1% [30]. The current research demonstrates that, as only two children (0.63%) were afflicted among 315 children, the risk of Klinefelter karyotype transmission is low [29]. Therefore, the present data reassures that KS men have no increased risk of transmitting their genetic problems to offspring.

### 2.8. Male with Robertsonian Translocation

During meiosis, carriers of Robertsonian translocation form trivalents. This results in alternate segregation, leading to normal or balanced sperm, and adjacent segregation, resulting in unbalanced sperm. Using the sperm-FISH method to analyze sperm chromosomes, approximately 73.6–91% of the sperm were reportedly normal or balanced [12,13], with the frequency of unbalanced sperm ranging from 3% to 36% [31]. The inter-chromosomal effect (ICE) has also been investigated, where translocated chromosome segregation may affect the segregation of other chromosomes. A review of six Robertsonian translocations showed higher rates of disomy and diploid sperm than controls [32]. A review of multiple cases by Anton et al. reported ICE in 54.5% of cases [33], with similar results reported previously. However, it is difficult to determine which effects result solely from ICE because Robertsonian translocation is associated with infertility and may be related to factors other than the ICE.

### 2.9. Males Carrying Reciprocal Translocations

During meiosis in males with reciprocally translated chromosomes, tetravalents are formed. Normal or balanced sperm are produced by alternate segregation, whereas unbalanced sperm are produced by adjacent type I, type II, and 3-to-1 segregation. The frequency of balanced and unbalanced sperm depends on the size of the translocation segment, number of recombination sites, and chromosome involved. Adjacent type I unbalanced sperm frequency ranges between 16% and 40%, whereas unbalanced sperm formed by adjacent type II and 3-to-1 segregation have 9% and 11% frequencies, respectively [10,31]. ICE has also been investigated in cases of reciprocal translocations. In a review of numerous cases, Anton et al. reported that 43.9% of cases involved ICE [33]. 

### 2.10. Males with Chromosomal Inversions

Chromosomal inversions have been linked to infertility and recurrent miscarriage, as reported in several studies [10]. A previous study used the sperm-FISH method to analyze multiple sperm and found that the frequency of recombinant chromosomal abnormalities in the sperm of males with chromosomal inversions ranged from 0% to 28% [34]. In addition, the size of the inverted chromosome segments was related to the frequency of chromosomal abnormalities [34]. The number of chromosomes with structural abnormalities was also reportedly associated with the size of the inverted chromosome segment.

Another study reported that the frequency of recombinant aberrant sperm in males with intra-arm chromosomal inversions ranged from 0% to 38% [33,35]. It also found that recombination-induced structural aberrations did not occur when the percentage of inverted chromosomes was less than 30% of the total. However, structural anomalies did occur at some frequency when the percentage was between 30% and 50%, and the frequency increased considerably when the percentage was ≥50%. Anton et al. reported a lower frequency (7.7%) of ICE than Robertsonian (54.5%) or reciprocal translocations (43.9%) [33].

### 2.11. Sperm Chromosome Testing and Clinical Application/Genetic Counseling

The clinical application of assisted reproductive technology (ART) techniques, such as in vitro fertilization (IVF) and ICSI, has enabled the conception of many couples in whom the male is infertile. These technologies use sperm from normally infertile men for insemination; however, there is still no established method for selecting only normal sperm through prior chromosome testing. Sperm from infertile men have a high rate of chromosomal abnormalities, resulting in a higher rate of chromosomal aberrations in fertilized eggs than in normal pregnancies.

Chromosomal examination of multiple spermatozoa using the sperm-FISH method has been conducted to predict the prognosis of ART pregnancies in infertile men. Sarrate et al. [18,19] reported that cases of oligozoospermia could be amenable to sperm chromosome analysis by the sperm-FISH method based on their study of various parameters in male infertility cases (Table 3). A 2014 review [36] reported that sperm chromosome analysis using sperm-FISH was performed in men from couples with normal semen parameters but repeated miscarriages and also in men from couples with normal semen parameters but repeated IVF failures. The results do not directly improve sperm quality or pregnancy outcome; however, they provide useful information for selecting treatment options for male infertility and selecting tests such as PGT-A after ART pregnancy, which can be applied to reproductive genetic counseling. According to a report compiling 18 years of sperm-FISH results, PGT-A may improve the cytogenetic results of fertile eggs in males in infertile couples with poor sperm-FISH results [18]. The authors also recommended the clinical application of sperm chromosome results obtained by sperm-FISH for genetic counseling. Other reports [37] have shown that an increased rate of sperm chromosomal aberrations detected using sperm-FISH leads to poorer PGT-A results. However, there have been reports [38] that applying PGT-A in patients with poor sperm-FISH results does not improve ART results, suggesting that the clinical significance of chromosome testing by sperm-FISH should be further investigated.

## 3. Oocyte

Unlike sperm, human oocytes have been extremely difficult to obtain for research purposes. However, with the advent of in vitro fertilization (IVF) embryo transfer, chromosomal analysis has been performed using surplus oocytes and polar bodies for preimplantation chromosome testing. Chromosome aberrations in embryos play a major role in the pregnancy rate and production rate of IVF embryo transfer. It is also known that most chromosome aberrations in embryos are aneuploidies and most causes of aneuploidy originate from the egg [39]. As women age, the occurrence of chromosomal aberrations increases as the egg ages. 

Sperm meiosis is continuous and short-lived; however, egg meiosis is unique because it begins before birth and progresses through a long dormant period until ovulation, where fertilization completes the meiotic cycle. The duration of meiosis in the egg corresponds to the female’s age at the time of ovulation. Differences in gametogenesis explain why the types and frequencies of chromosomal aberrations in oocytes differ significantly from those in sperm.

Human oocyte chromosome analysis is based on a single cell and requires a more reliable and accurate chromosome analysis method than methods that focus on many cells, such as lymphocytes. Recent developments in molecular biology techniques have led to significant differences from results obtained using classical analytical methods. In addition, the specimens used for analysis are greatly influenced by the state of the target oocytes, such as surplus oocytes, unfertilized eggs, or polar bodies of oocytes used for in vitro fertilization-embryo transfer, which cause significant differences in the results [40]. 

This overview provides insights into the mechanisms underlying oocyte formation and abnormal development, history and methods of chromosome analysis, and results of chromosomal aberrations [40].

### 3.1. Meiosis of the Oocyte

In oogenesis, the process proceeds to the diplotene stage of the meiosis prophase during the embryonic period and then stops until resumption after puberty. Ovulation of the oocyte between puberty and menopause completes the first meiotic division, and fertilization completes the second meiotic division. The aneuploidy here is often generated during meiosis I and has been largely explained by the non-disjunction model. However, with the development of molecular genetic analysis methods, other modes of segregation that cause aneuploidy have been revealed. The oocytes in the metaphase of meiosis II are generally called secondary oocytes. After that, sperm entry into the egg (fertilization) releases the second polar body, completing meiosis II. The duration of meiosis in the egg corresponds to the age of the female during ovulation. 

Meiosis is characterized by the uniqueness of meiosis I. All homologous chromosomes doubled from the oocyte are linked by recombinant chiasmata, and the two sister chromatids that constitute the chromosome are linked by cohesin, a protein complex. Homologous chromosomes bound via chiasmata during meiosis I are aligned in the equatorial plane due to the tension of spindle threads bound from both poles to the centromere (kinetochore). The homologous chromosomes that are typically aligned in the equatorial plane are then separated into two poles, and one set is released from the egg cell as the first polar body owing to the loss of cohesin at sites other than the centromere [2,40].

Conversely, meiosis II follows the same mode of segregation as somatic cell division by separating the two sister chromatids in anaphase and releasing them as the second polar body. Anaphase of meiosis II is initiated by fertilization of the sperm and oocyte.

### 3.2. The Mechanism Underlying Abnormal Development by Meiosis in Oocytes

Chromosomal aberrations in oocytes are believed to occur primarily during meiosis I. Aneuploidy in meiosis I is thought to be caused by chromosome non-disjunction, early chromosome non-disjunction (predivision) [41], and late delay. However, early chromosome segregation is considered to be the primary cause. This is due to factors such as the absence of chiasma formation in homologous chromosomes, cohesin depletion associated with aging [42], and abnormalities in the spindle [43] and centromere. 

Chiasmata are formed randomly, and they tend to be more numerous on long chromosomes than on short ones. When homologous chromosomes lack chiasmata, they tend to separate early in the middle of meiosis I, which results in unusual forms of segregation in meiosis I, including the separation of sister chromatids that originally occurred during meiosis II.

Cohesins are crucial in maintaining the two sister chromatids during DNA replica-tion. Cohesin is not cleaved, and homologous chromosomes are combined until late in meiosis I; however, the number of cohesins decreases with age, resulting in the premature separation of homologous chromosomes. Moreover, chiasmata are more likely to be lost when they are present at the ends of chromosomes due to a decrease in cohesion, which makes chromosomal aneuploidy more common in short chromosomes than in long chromosomes. Cohesin depletion is associated with aging.

The aneuploidy rate does not change substantially until the age of 35; it gradually increases after 35 years and sharply after 40 years. Various factors are involved; however, a decrease in cohesin levels is considered the most significant cause.

### 3.3. Oocyte Chromosome Analysis Methods

The analysis of oocyte chromosomes requires a technique to examine the chromosomes of an individual cell. Since the oocyte is in meiosis II, the chromosomes can only be observed through direct fixation. Meiosis II chromosomes do not get G-banded, which is the most common approach for chromosome identification. Therefore, routine Giemsa staining makes karyotyping and chromosome identification of meiotic metaphases challenging. A reliable and accurate specimen preparation method is necessary for proper preparation. Tarkowski’s chromosome preparation method, which is simple and easy, tends to rupture the cell membrane and scatter chromosomes due to its rapid fixation. As a result, the percentage of hypoaneuploidy, which affects a small number of chromosomes, was higher than that of hyper aneuploidy. To address this issue, Mikamo and Kamiguchi [44] developed an advanced fixation-air-drying method that gradually fixes the egg cell membrane using a three-step fixative solution to prevent membrane rupture and reduce chromosome dispersal. Furthermore, most chromosomes were analyzed using Giemsa staining (Figure 1).

In recent years, chromosomal analysis methods have been improved using molecular genetic techniques. The FISH method can identify chromosome-specific sites and analyze meiosis metaphase II and polar body chromosomes. However, the number of chromosomes that can be analyzed using FISH is limited to approximately 3–5 because a limited number of different fluorescent labels can be used simultaneously. There are accuracy issues due to variations in the analysis results caused by the condition of the fixed specimens. Furthermore, whole genome amplification (WGA) of a single cell can analyze the aneuploidy and structural variation of all chromosomes using array comparative genomic hybridization (aCGH) and next-generation sequencing (NGS) methods. Whole-chromosome aneuploidy and structural aberrations can now be analyzed. NGS is currently the primary method used in preimplantation testing for the chromosomal analysis of embryos. This method separates the oocyte from the polar body and allows chromosomal analysis of each WGA product [45].

### 3.4. Chromosome Abnormalities in Oocytes

Chromosomal analysis of the secondary oocyte provides information on the abnormalities resulting from the first meiotic division. The Tarkowski method has been used in various studies, including one by Tejada et al. [46] who reported a chromosome aberration rate of 28.7% in 334 analyzable oocytes from women aged 23–40 years. Of these, 19.2% were aneuploid (9.3% hypoaneuploidy and 9.9% hyperaneuploid), and 3.9% were diploid. In other reports [47,48,49,50] analyzing large numbers of oocytes, the chromosome aberration rate ranged from 27.1% to 46.9%, with an average of 34.9%.

In Pellestor’s study [51], the largest number of oocytes was analyzed using the progressively fixed air-drying method developed by Mikamo and Kamiguchi. This study analyzed 1397 oocytes and found a chromosome aberration rate of 22.1%, aneuploidy of 10.8% (hypoaneuploidy 5.4%, hyperaneuploidy 4.1%, combined + many aneuploidies: 1.3%), 5.4% diploidy, 3.8% sister chromatids alone, and 2.1% structural aneuploidy. They also discovered that of the 132 aneuploidies, 62.9% were chromosome-type (meiosis I) aneuploidies, which were more common than chromatid-type (meiosis II) aneuploidies. The average rates observed in these studies [52,53,54,55] indicated a chromosome aberration rate of 21.8%, aneuploidy of 11.1% (hypoaneuploidy: 6.1%, hyper aneuploidy: 4.5%), ploidy of 7.9%, and structural aberration of 2.9%. In addition, Pellestor et al. [56] showed an association between female age and chromosome aberration rate (Figure 2).

Chromosomal analysis using fluorescence in situ hybridization (FISH) analyzes only two to nine chromosomes. In chromosome analysis using metaphase oocytes, there have been varying aneuploidy rates reported. Dyban et al. [57] reported a high aneuploidy rate of 36.8% in 156 oocytes with chromosome 18 and X probes. Anahory et al. [58] reported a high aneuploidy rate of 47.5% in 54 oocytes with eight probes. In contrast, Honda et al. [59] reported an aneuploidy rate of 3% for 183 oocytes using probes targeting the 18, 21, and X chromosomes. However, Cupisti et al. [60] reported an aneuploidy rate of 3.8% for 203 oocytes using eight probes. In the chromosome analysis of polar bodies using the FISH method, Verlinski et al. [61] found an aneuploidy rate of 43.1% in 3217 oocytes (aneuploidy in the first polar body: 35.8%, aneuploidy in the second polar body: 26.1%) using three different probes. In contrast, Kuliev et al. [62] found an aneuploidy rate of 52.1% in 6733 oocytes using five different probes (anomalies of the first polar body: 41.7%, aberrations of the second polar body: 30.7%). The differences in aneuploidy rates among studies may be due to the characteristics of the FISH analysis method, resulting in large variations in the data.

Gabriel et al. [63] conducted polar body analysis using the array CGH method to distinguish between whole-chromosome (non-disjunction) and chromatid (precocious separation) errors. They found that of 164 human first polar bodies, 86 (47.6%) had aneuploidy with no difference between low (55.3%) and high aneuploidy (44.7%). However, hypoaneuploidy was twice as common as hyper aneuploidy for chromosomal aberrations. Chromatid errors accounted for 92% of all aneuploidies and were more than 10 times more frequent than whole chromosomal errors (8%).

Fragouli et al. [64] analyzed the chromosomes of 420 oocyte polar bodies using the aCGH method. At an average age of 40.7 years, they found that 26% of women were normal, 30% had abnormalities in the first polar body, 43% had abnormalities in the second polar body, and 27% had abnormalities in both polar bodies. The study found that abnormalities in meiosis II were more common than those in meiosis I. Moreover, as female age increased, aneuploidy also increased, with 47% in the younger group (mean age, 36 years) and 78% in the older group (mean age, 41 years). Furthermore, younger women had more anomalies in meiosis I, while older women had more anomalies in meiosis II.

Verpoest et al. [65] analyzed 1023 first and second polar bodies obtained from 205 individuals with an average age of 38.6 years. Chromosomes were analyzed using the array CGH method, and the results showed that 525 (64.7%) of the 811 oocytes available for analysis showed aneuploidy.

In addition, Capalbo et al. [66] reported the chromosome analysis of polar bodies, cleavage-stage embryos, and blastocysts of 21 oocytes from older women over 40 years of age with good ovarian function. All oocytes were chromosomally abnormal, with 91.7% (22/24) of the abnormal chromosomes in the first polar body containing aberrant chromosomal segments.

## 4. Embryo

Many fertilized human eggs (embryos) exhibit chromosomal abnormalities that often result in failed births due to embryonic arrest, implantation failure, miscarriage, or intrauterine fetal death. The estimated rate of chromosomal abnormalities among all newborn babies is 0.7%. Most chromosomal aberrations observed in embryos are aneuploidies, but structural anomalies, polyploidy, and mosaic embryos are also observed [67]. Aneuploidy in embryos is caused primarily by meiosis in eggs, whereas structural anomalies are mainly caused by sperm. Aneuploidy is closely associated with female aging and a major cause of decreased pregnancy and increased miscarriage rates in older women. Rates of chromosomal aberration in blastocysts, which are 5-day embryos, are found in approximately 1/3 of women aged 35 years or younger, 1/2 at age 38, and 2/3 at age 41 (Figure 3) [68]. Since morphological evaluation alone cannot diagnose the presence of chromosomal abnormalities, preimplantation diagnosis (PGT-A), which examines the chromosomes in the embryo, is often performed.

Yang et al. [69] reported that of 44.9% of blastocysts with chromosomal aberrations, 35% had monosomy, 21% had trisomy, 29% had two chromosomal abnormalities, and 15% had three or more chromosomal aberrations. Embryos with monosomy and multiple chromosomal aberrations typically cannot develop into blastocysts, but few aborted fetuses can implant successfully.

In addition to the aneuploidy and structural abnormalities caused by the fertilization of abnormal gametes, embryos can also develop polyploidy (triploidy or tetraploidy), which is caused by multi-sperm fertilization at the time of fertilization, failure of the second polar body release of the egg, abnormal first ovarian split, and mosaicism, which is caused by abnormal somatic cell division after fertilization [66]. To test for these abnormalities, Gruhn et al. [67] compared the analysis of the trophectoderm (TE) in a blastocyst on day five with that of cleavage-stage embryos on day 3 after fertilization as a preimplantation test. They found that the percentage of chromosomal aneuploidy when cleavage-stage embryos were examined was higher than that when blastocysts were examined, indicating that the chromosomal aberration rate of embryos was higher when embryos were examined at an earlier stage. This may be because embryos with abnormal karyotypes stop cell division in midstream and do not develop. Furthermore, the chromosome aberration rate of embryos is negatively correlated with age from 20 to 27 years but increases after 27 years of age [67]. Oocytes from young women under 20 years of age tend to present higher rates of chromosome segregation errors than those from women in their 20s and early 30s [2]. In the first years of ovulation, women under 20 years of age experience high rates of chromosome segregation errors where the entire homologous chromosomes mis-segregate during meiosis I. However, the mechanism underlying why aneuploidy is more common in the oocytes of young women is unclear, thus warranting further study.

Unlike aneuploidy, mosaicism is not associated with female age, and external factors such as in vitro culture are believed to be involved [70]. The frequency of mosaicism is reportedly 30% in dividing embryos (early embryos) and 10–30% in blastocysts, although there are differences among reports. Mosaicism is found in a small percentage of miscarriages. Aneuploidies are uncommon in live births, except for trisomies 21, 18, and 13, and sex chromosome aneuploidies, which are often viable. Aneuploidy in the human blastocyst primarily originates from maternal meiosis; only 2% display evidence of mitotic-origin aneuploidy. Furthermore, a more rigorous classification of mosaicism through cell-division origin-of-aneuploidy analyses might ultimately improve our understanding of the true reproductive potential of bona fide mosaic embryos [71].

## 5. Conclusions

We have reviewed the results of sperm chromosome analysis and noted that sperm-FISH had enabled the analysis of a large number of sperm, leading to numerous clinical trials on aneuploid sperm in men with infertility and chromosomal abnormalities. However, a thorough assessment of individual sperm, including structural abnormalities, requires using hamster and mouse oocytes. It is also desirable to develop a method for the cytogenetic evaluation of sperm for ICSI. Thus far, clinical trials have demonstrated that the rate of sperm chromosomal aberrations is increasing in infertile men. A decrease in sperm concentration is most closely related to the rate of sperm chromosomal aberrations. Men with chromosomal aberrations have a higher rate of sperm chromosomal abnormalities. A high percentage of oocytes exhibit chromosomal abnormalities, most of which are aneuploidies. This considerably differs from 90% of sperm chromosome aberrations observed in approximately 15% of cases, which are structural aberrations [55]. Aneuploidy of oocytes is more common than aneuploidy of sperm due to early chromosome segregation during meiosis I. It is a vital cause of the increasing rate of chromosomal aberrations with increasing female age. In addition, the application of PGT-A has shown promising results in cases with a high rate of sperm chromosome aberrations observed using sperm-FISH. The clinical application of PGT-A in genetic counseling requires further investigation.

## Figures and Tables

**Figure 1 biomolecules-13-01010-f001:**
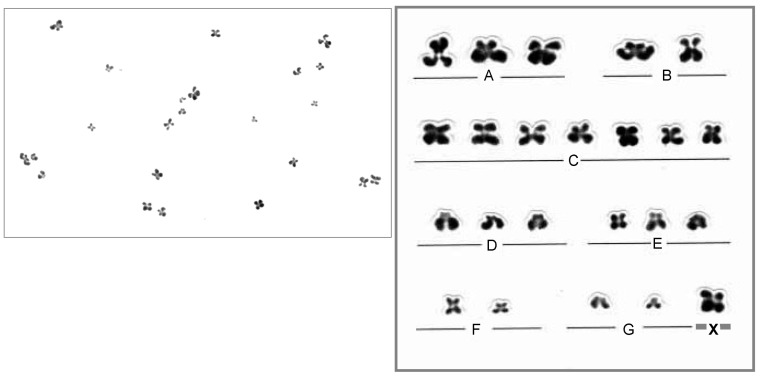
Karyotype of oocyte using Giemsa staining. A, B, C, D, E, F, G and X indicate the respective chromosome groups in figure caption.

**Figure 2 biomolecules-13-01010-f002:**
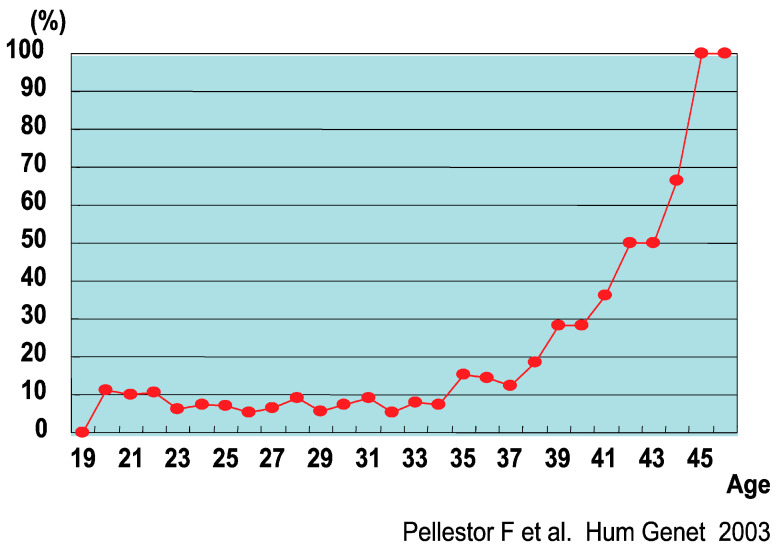
Association between female age and chromosome aberration rate of oocytes [56].

**Figure 3 biomolecules-13-01010-f003:**
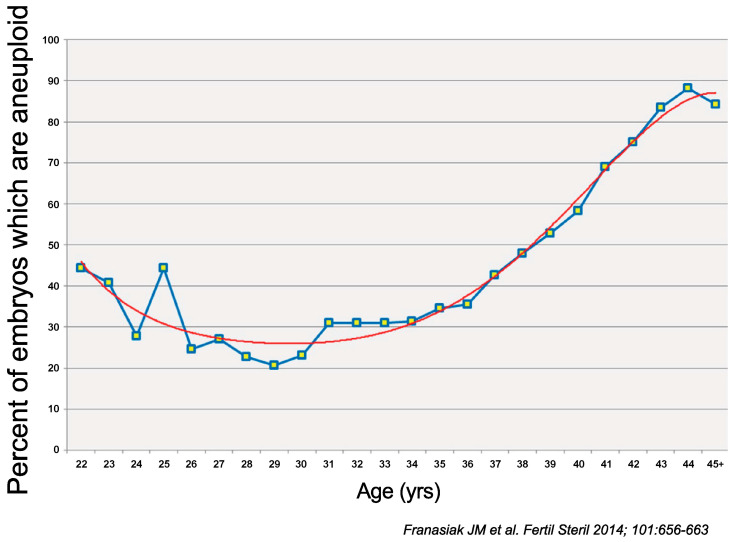
The prevalence of aneuploidy relative to the female partner’s age demonstrates the lowest risk in women from their middle to late twenties, with significantly higher rates in embryos obtained from both younger and older women (*p* < 1 × 10^−6^). The relationship between age and the aneuploidy rate best fits the 5th – degree polynomial (regression line shown) [68].

**Table 1 biomolecules-13-01010-t001:** Comparison of Sperm Chromosome Analysis.

Method	Pros	Cons
Zona-free hamster eggs	Can be analyzed by karyotype	Technically difficult Only swim-up sperm can be analyzed Difficult to analyze multiple sperm
ICSI of mouse egg	Can be analyzed by karyotype	Difficult to analyze multiple sperm
Sperm-FISH	Capable of analyzing multiple sperm	Difficult to analyze structural anomalies Signal analysis standards and accuracy vary between laboratories

**Table 2 biomolecules-13-01010-t002:** Sperm aberration rate of men with sex chromosome aneuploidy (median ± SD) [12].

Karyotype	XX Disomy (%)	YY Disomy (%)	XY Disomy (%)
47, XYY	1.65 ± 2.31	1.54 ± 1.53	4.43 ± 6.03
47, XYY/46, XY	0.17 ± 0.16	0.50 ± 0.45	0.48 ± 0.31
47, XXY	4.64 ± 2.56	0.30 ± 0.46	11.1 ± 6.89
47, XXY/46, XY	0.40 ± 0.31	0.42 ± 0.49	1.22 ± 0.71

**Table 3 biomolecules-13-01010-t003:** Patients with abnormal FISH results classified according to their seminal parameters.

Seminal Parameters	Altered FISH Results	%
Astheteratozoospermic	6/71	8.5
Asthenozoospermic	4/67	6.0
Normozoospermic	4/34	11.8
Oligoasthenoteratozoospermic	13/62	21.0
Oligoasthenozoospermic	17/51	33.3
Oligoteratozoospermic	2/13	15.4
Oligozoospermic	2/4	50.0
Teratozoospermic	1/17	5.9
Total	49/319	15.36

Note: FISH: fluorescence in situ hybridization. Sarrate et al. Sperm FISH studies. Fertil Steril 2010.

## Data Availability

Not applicable.

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
