# Peer review of "Sperm and Oocyte Chromosomal Abnormalities"

_biomolecules, 2023, doi:10.3390/biom13061010_

Round 1

Reviewer 1 Report

In this review, the authors summarized the investigations and findings of chromosomal aberrations in human sperm, oocytes, and embryos. I would like to offer some comments:

1. The majority of the literature appears to be outdated, indicating that the subject matter is not currently a significant area of research. It is recommended that the authors incorporate more contemporary and elegant studies, as well as in-depth investigations into the underlying mechanisms, in order to reinforce their opinions.

2. The opinions of ICE (p. 4 and 5) may require additional substantiation to be considered definitive, as the current evidence is not entirely conclusive.

3. It is unlikely that FISH has the capability to identify chromosomal structural abnormalities in sperm (line 48). FISH can only detect aneuploidy in sperm nuclei.

Author Response

Point-by-point responses to comments from Reviewer 1

 We would like to thank Reviewer 1 for your time and efforts in reviewing our manuscript and for providing comments, which have helped us improve our manuscript considerably. We have made revisions based on your comments and have provided our point-by-point responses below. We hope that our responses and revisions appropriately address your comments. 

Comment 1: The majority of the literature appears to be outdated, indicating that the subject matter is not currently a significant area of research. It is recommended that the authors incorporate more contemporary and elegant studies, as well as in-depth investigations into the underlying mechanisms, in order to reinforce their opinions. 

Response to Comment 1:

Thank you for this helpful feedback. According to the Reviewer’s suggestion, we have added the following information to the introduction. We have added 6 new references to this paper (reference # 1, 2, 27, 29, 20,71):

Most chromosomal aberrations are known to occur during gametogenesis, especially in meiosis. Gametogenesis begins in the primordial germ cell, which undergoes repeated somatic cell divisions, and finally enters meiosis with replicated DNA. The early phase of meiosis I is classified into the leptotene, zygotene, pachytene, diplotene, and diphyletic phases. During this process, chromosomes undergo a homology search, and homologous chromosomes pair to form the synaptonemal complex. Also, recombination between homologous chromosomes to repair the DNA double-strand breaks (DSBs) on each chromosome caused by Spo11 or other proteins results in the formation of chiasmata. These are important structures for creating diversity and subsequent accurate chromosome segregation. It is known that most aneuploidy identified in the embryonic, perinatal, and neonatal stages occurs during oogenesis.” (Lines 28–38)

 Comment 2: The opinions of ICE (p. 4 and 5) may require additional substantiation to be considered definitive, as the current evidence is not entirely conclusive. 

Response to Comment 2:

Thank you for this comment. According to the Reviewer’s suggestion, we have added the following part in the introduction and oocyte.

However, it is difficult to determine which effects result solely from ICE, because Robertsonian translocation is associated with infertility and may be related to factors other than the ICE..(  Line 535-537)

 Comment 3: It is unlikely that FISH has the capability to identify chromosomal structural abnormalities in sperm (line 48). FISH can only detect aneuploidy in sperm nuclei. 

Response to Comment 3:This helpful comment is appreciated. Following your suggestion, we have revised the relevant sentence as follows (line111): “…making it possible to analyze aneuploidy in a large number of sperm.” 

Reviewer 2 Report

Comments

While the topic attracted interest, reading the manuscript was a disappointment because of incorrect use of basic terminology that applies to gamete cytogenetics and meiosis. The authors have also failed to highlight the main challenges in clinical cytogenetics of sperm and present sperm analysis by FISH as any other FISH analysis, which is misleading. Curiously, after presenting the rates of cytogenetic abnormalities in the sperm of infertile men and those with known chromosomal aberrations (such as XXY), the authors do not even comment on the fact that the use of assisted reproduction technologies in these cases will propagate these aberrations to the next generation. In my opinion, the manuscript requires thorough revision for contents, scientific terminology, and English (many unnecessary hyphens, wrong punctuation, odd syntax). Also, the referenced literature is relatively old – out of 65 references only 24 are published after 2010.

Specific comments

Abstract

-          Line 10: “Most oocyte chromosomes are aneuploid,….” reads odd. Should be like “ Most oocytes have chromosomal aneuploidies, …”.

1. Introduction

- Introduction is very shallow and should be more elaborated.

-          It is not correct to write about “two meiosis cycles”. Meiosis is one process with two consecutive cell divisions MI and MII.

-          “Gametophyte” is a term from botany and not appropriate for humans. The result of meiosis are HAPLOID (not monoploid) GAMETES.

-          I think it is incorrect writing that “most oocyte chromosomes are aneuploid”. Aneuploidy due to mistakes in MI or MII segregation happens but it usually affects one or two chromosomes, not all. See comment for the Abstract

2. Sperm

-          While it is true that sperm chromosomes cannot be analyzed the same way as blood lymphocyte chromosomes, the main reason is that sperm are terminally differentiated cells and do not undergo cell division during which chromosomes are best visible. For correctness, blood lymphocytes also do not naturally go through cell division – they are stimulated by mitogens into mitosis. Please reword.

-          “Clear zone-removed hamster egg” should be “zona-free hamster egg”.

-          Line 41: what is “nuclear plate”. Do the authors mean “metaphase plate”?

-          Line 45: Please give full name of the acronym ICSI at its first appearance.

-          Line 50: “Sperm” and not “sperms” is a plural and should be used as such. Singular would be “sperm cell” or more correctly “spermatozoon”.

Table 1

-          Column one: Should be “zona-free hamster eggs.

II Results of sperm chromosome analysis

-          When writing about analyses using hamster eggs, it should be written “zona-free hamster egg”, so it is distinct from mouse egg analysis by ICSI. Please correct throughout the text.

-          Line 58: the sentence about DNA damage repair gives a false impression as if sperm of non-human mammals is capable for DNA repair. Please reword.

-          Line 61: Please replace “motility-positive sperm” with “motile sperm”.

-          Line 63: Please replace “This result aligns” with “These results are consistent with”.

III Chromosome analysis results from FISH-based analysis

-          Please reword the title of this section as “Sperm chromosome analysis by sperm-FISH” and please use the term “sperm-FISH” instead of just “FISH” when writing about sperm chromosome analysis by FISH. This is because technically, sperm-FISH is different from conventional chromosome FISH. I suggest adding a small paragraph about sperm-FISH technique.

-          Please explain why many studies have used probes for 13, 18, 21, X and Y only. There is a good reason for that.

-          While writing about sperm disomy and nullisomy, is there information about the rate of disomy or nullisomy for multiple chromosomes in the same sperm?

-          Lines 75-77: Is the rate of autosomal and sex chromosome disomy/monosomy the same?

-          Lines 81-83: the factors contributing to male infertility should be “ low sperm count, low motility, and abnormal morphology”.

-          Line 83: should be “oligoasthenoTERATOzoospermia”.

-          Line84-85: should be “where sperm count, motility and morphology are poor…”

-          Line 85: nonobstructive azoospermia is when NO SPERM are collected from testes (impaired spermatogenesis).

-          Line 86: should be “abnormality rates have also been ……”.

-          Lines 88-89: should be “…, and frequency of ABNORMAL sperm morphology.”

-          Lines 90-92: the sentence about mouse oocytes reads weird. What are “other mice”? Which reference?

-          Line 101: should be “remains”.

-          Line 105: should be “examined”.

-          Line 106: “several telomere probes” reads odd because telomeric sequences are the same in all chromosomes. Please reword.

-          Line 109: should be “disomic and diploid sperm”. Did the authors, indeed, mean DIPLOID? This has not been mentioned earlier where the authors write only about DISOMIC and MONOSOMIC sperm.

-          Lines 116-117: should be “have an extra Y chromosome as a result of sister chromatid nondisjunction in Meiosis II or MII”.

-          Line 117: should be “the extra Y chromosome is…” and this sentence needs a reference. Or….do the authors mean that both Y chromosomes are lost? In such case, there should be sperm with no Y resulting in X-monosomy in a zygote. Please clarify.

-          Line 118: remove “Despite this” as it makes no sense.

-          The whole paragraph about 47,XYY males does not read well. Earlier, the authors write that sperm chromosome analysis has studied all chromosomes now that it is mainly about sex chromosomes???? The last sentence about mosaic cases does not make sense: no difference in frequency compared to what?

-          Line 140: should be “aneuploidy”.

-          Lines 145, 152, 155: should be “Robertsonian”

-          Line 145: should be “trivalents” (remove “chromosomes”).

-          Line 149: use “unbalanced” instead of “disproportionate”.

-          Line 152: here the authors use “spermatozoa” instead of “sperm”. Please be consistent in using one or another.

-          Line 157: please reword the subheading as “Males carrying reciprocal translocations”.

-          Line 158: should be “tetravalents” (remove “chromosomes”.

-          Line 160: should be “unbalanced” or “genetically unbalanced” (remove “disproportionate”).

-          Line 162: Please reword as “….the number of recombination sites, and the specific chromosomes involved”.

-          Line 167: reword subtitle as “Males with chromosomal inversions”.

-          Line 168: change “Inverted chromosomes” to “Chromosomal inversions”.

-          Lines 170-171: reword as “…in the sperm of males with chromosomal inversions, ranged …..”.

-          Lines 175-176: This sentence should read as “ Another study reported that the frequency of recombinant aberrant sperm in males with intra-arm chromosomal inversions ranged from 0 to 38%”.

-          Line 177: should be chromosomal inversions not “inverted chromosomes”

IV Sperm chromosome testing ….

-          Line 195: should be “…that sperm chromosome analysis by FISH was …”

-          Lines 201-202: should be “ compiling 18 years of sperm-FISH results, …”

-          Line 203: should be “….with poor sperm-FISH results”.

-          Table 3: correct spelling is “oligoasthenoteratozoospermic

-          Line 212: should be “fluorescence”

3. Oocyte

-          The text should be in Present (it is) or Present Perfect (it has been) Tense and not in the Past Tense (it was). Please correct where appropriate.

-          Line 217: should be “sperm” not “sperms”

-          Line 222: should be “aneuploidies”

-          Please elaborate the timeline of female meiosis providing the time during embryonic development when meiosis starts, at which stage it stops, etc.

-          Lines 245-247: This is not correct that during ovulation just one oocyte resumes meiosis. Typically, it is a group of oocytes.

-          Line 246: Acronyms LH and HCG need explanation.

-          Lines252-260: Please consult literature for proper description of Meiosis I. For example, there is no mentioning of synaptonemal complex proteins and synapsis. What are “bipolar chromosomes”?  Please use the proper terms for the specific stages (metaphase, anaphase, etc.) of meiosis I (MI) and meiosis II (MII) instead of ambiguous terms as ‘mid-meiotic chromosomes”. The last sentence of this paragraph is long and confusing. Please reword for clarity.

-          Line 268: should be “chromosome non-disjunction”.

-          Line 273: Plural for “CHIASMA” is “CHIASMATA”. The authors use this correctly in some sentences but not in others.

-          Line 279: “fixed” is not a proper term in this context.

-          While talking about oocyte chromosome analysis methods, it should be mentioned that MII chromosomes do not get G-banded, which otherwise is the most common approach for chromosome identification. Therefore, routine Giemsa staining is used making karyotyping and chromosome identification of meiotic metaphases challenging.

-          Line 304: please use “meiosis metaphase II chromosomes” instead of “mid-meiotic chromosomes”; should be “polar body chromosomes”. And from where come interphase nuclei? Female meiosis gives just one oocyte and one polar body.

-          Please explain why the number of chromosomes that can be analyzed by FISH is limited.

-          Line 315: should be “secondary oocyte”.

-          Lines 318-320: “…19.2% were aneuploid….., 3.9% were aneuploid” does not make sense. The second should likely be “diploid”. And what is “monogenetic”? Please use proper terminology.

-          Lines 327-329: Please clarify the meaning of chromosome-type aneuploidies versus chromatid-type aneuploidies. Is this referring to mistakes in MI and MII, respectively?

-          Lines 351-358: This paragraph is difficult to understand. What does “164 chromosomes” mean? What is “aneuploidy for the combined chromosome segment”? What is “chromosome body type aneuploidy”. What are “hypo heterogeneity” and “hyper heterogeneity”? Again, what are “chromatid type anomalies” versus “chromosomal anomalies?” I am quite familiar with cytogenetic terminology but not these ones.

4. Embryo

-          Line 381: The sentence “The estimated rate of such failure is 0.7% of all births.” is confusing. How can embryonic arrest and implantation failure be calculated from all births? Could it be “pregnancies” instead of “births”? But then again, 0.7% is too low for all pregnancies. Please clarify.

-          Line 382: should be “aneuploidies”; what is “ploidy”, did the authors mean “polyploidy”. Please correct.

-          Line 387: remove “fertilized”.

-          Line 400: remove “duplicate”.

-          Line 401: remove “multiple”.

-          Line 405: replace “aneuploidy” with “polyploidy (triploidy or tetraploidy)”

-          Line 407: what is “first ovarian split”?

-          Line 409: should be “trophoectoderm”.

-          The more commonly used term for “dividing embryos” are “cleavage stage embryos”. Consider changing throughout the text.

-          Line 423: should be “…., and sex chromosomes which are often viable.

5. Conclusion

-          Lines 435-436: sentence about the interchromosomal effects reads odd. Please reword.

Lines 439-440: “Aneuploidy is more common than aneuploidy…..” reads odd, the sentence is missing some

Round 2

Reviewer 1 Report

No other comments.

Author Response

We wish to re-submit the manuscript titled “Sperm and Oocyte Chromosomal Abnormalities.” The manuscript ID is 2322822.

We thank you and the reviewers for your thoughtful suggestions and insights. The manuscript has benefited from these insightful suggestions. I look forward to working with you and the reviewers to move this manuscript closer to publication in Biomolecules.

Reviewer 2 Report

Comments

The authors have made some of the requested revisions but failed to make others. In addition, while making suggested changes, they have generated additional errors or inaccuracies. English has improved but needs additional editing. Please also check text for typos, punctuation, and spaces. The word ”furthermore” has been overused.

Specific Comments

Abstract

-          Line 9: remove “are produced”.

-          Line 10: The authors have failed to make the requested change. They write incorrectly that “Notably, most chromosomes have chromosomal aneuploidies.” Instead, it should be that Most oocytes have chromosomal aneuploidies, …”. Please correct again.

-          Line 13: should be “increased NUMBER of chromosomal aberrations.”

-          Line 21: should be “..that HAVE BEEN reported..”

1. INTRODUCTION

Lines 30-32: As suggested, the authors have added new text but this brings in additional inaccuracies: The early phase of Meiosis I is called PROPHASE, which is divided into five stages (these must be written as Nouns and not as Adjectives): LeptoNEMA, ZygoNEMA, PachyNEMA, DiploNEMA, and DIAKINESIS.

Line 36: should be “to create GENETIC diversity…”

Line 37: should be “ANEUPLOIDIES…..OCCUR”

Lines 37-39: The sentences starting “It is known….” and “Notably, most…” are essentially redundant. Please reword.

2. Sperm

Line 50: should be “zonA-free”

Line 51: should be “METAPHASE”

Line 52: should be “ established this AS a revised method”

Line 60: should be “multiple sperm” because SPERM is plural

Table 1 last column lower: should be: “Signal analysis standards and accuracy vary between laboratories

Line 79: replace “normal blood cells” with “normal SOMATIC cells”

Line 80: should be “difficult for probes to hybridize to sperm chromatin”

Line 81: should be: “necessary to DECONDENSE sperm nuclei..”

2.4 lines 87-95 does not read good. The two first sentences are redundant. Also, the authors failed to explain the reason why most researches have focused on aneuploidies of human chromosomes 13, 18, 21 and the sex chromosomes. It is not correct that these chromosomes have higher frequency of aneuploidies. It is just that trisomies for these three autosomes and aneuploidies of sex chromosomes are more viable than aneuploidies of other autosomes. Please revise the entire paragraph.

Line 112: should be “the number of sperm with disomy for chromosomes 13,…”

2.8 should be “Males with Robertsonian translocations”

Line 180: should be “UNBALANCED”

3. Oocyte

Line 279: replace “of the first premeiotic division” with “of meiosis prophase”

Line 285: not clear what is the meaning of the sentence “The oocytes in the metaphase of meiosis II are generally called oocytes.” Something is missing, “secondary oocyte”?

Line 295: replace “two ends” with “two poles”

Line 299: sister chromatids are separated in anaphase, not metaphase

Lines303-304: the sentence “Aneuploidy….” does not make sense. Disjunction is the normal process in meiosis I  should not cause any aneuploidies. Aneuploidies are caused by nondisjunction. Please reword.

Lines 111-112: should be “occur”

Lines 320-322: change the Tense in this paragraph from Past to Present.

Line 340: replace “implemented” with “improved”

Line 344: change to “because a limited number of different fluorescent labels can be used simultaneously”

Line 389: change as “Gabriel et al. CONDUCTEDpolar body analysis…”

4. Embryo

Line 421: remove either “mainly” or “primarily”, redundant

Line 459: the sentence “It is uncommon…” reads odd. Change to “Aneuploidies are uncommon in live births, except for trisomies 21, 18, and 13, and sex chromosome aneuploidies, which are often viable”

5. Conclusion

Line 468: should be “SPERM” which is plural.

Line 471: should be “HAMSTER” not “hamsters”

Line 475: remove “in semen parameters”

Lines 475-476: the sentence “Men with chromosomal aberrations have a higher rate of chromosomal aberrations” does not make any sense.

Line 477: should be “…most of which are ANEUPLOIDIES”

Finally, the authors have explained why elder women have higher rate of aneuploidies. However, there is no explanation why younger women also have higher rate of aneuploidies?

Author Response

Point-by-point responses to comments from Reviewer 2

 We would like to thank Reviewer 2 for the time and effort in reviewing our manuscript and for providing comments, which helped us improve our manuscript considerably. We have made revisions based on these comments and have provided point-by-point responses below. We hope our responses and revisions appropriately address the comments raised by the Reviewer.

Reviewer #2’s General Comment

The authors have made some of the requested revisions but failed to make others. In addition, while making suggested changes, they have generated additional errors or inaccuracies. English has improved but needs additional editing. Please also check text for typos, punctuation, and spaces. The word ” furthermore” has been overused.

Response to General Comment

Thank you for this helpful feedback. We have made revisions according to the Reviewer’s suggestions throughout the manuscript. We have carefully reviewed and revised the manuscript to correct the language, grammar, and formatting of terminology. We have addressed each comment both in the rebuttal letter and in the revised manuscript.

Specific Comments

Comment 1: ABSTRACT

Line 9: remove “are produced”.

Line 10: The authors have failed to make the requested change.  They write incorrectly that “Notably, most chromosomes have chromosomal aneuploidies.” Instead, it should be that “MOST OOCYTES have chromosomal aneuploidies, …”. Please correct again.

Line 13: should be “increased NUMBER of chromosomal aberrations.”

Line 21: should be “..that HAVE BEEN reported..”

Response to Comment 1:

Thank you for this helpful suggestion. Accordingly, we have made the appropriate revisions in the Abstract.

Comment 2: 1. INTRODUCTION

Lines 30-32: As suggested, the authors have added new text but this brings in additional inaccuracies: The early phase of Meiosis I is called PROPHASE, which is divided into five stages (these must be written as Nouns and not as Adjectives): LeptoNEMA, ZygoNEMA, PachyNEMA, DiploNEMA, and DIAKINESIS.

Line 36: should be “to create GENETIC diversity…”

 Response to Comment 2:

We have made revisions according to the Reviewer’s suggestions.

Comment 3:

Line 37: should be “ANEUPLOIDIES…..OCCUR”

Lines 37-39: The sentences starting “It is known….” and “Notably, most…” are essentially redundant. Please reword.

 Response to Comment 3:

Thank you for this helpful feedback. Incorporating these suggestions, we have made the following revisions:

It is known that most aneuploidy identified in the embryonic, perinatal, and neonatal stages occurs during oogenesis.  Most aneuploidies are of oocyte origin, as indicated by the origin of chromosomal aberrations in miscarried and newborn infants [2]

Comment 4: SPERM 

Line 50: should be “zonA-free”

Line 51: should be “METAPHASE”

Line 52: should be “ established this AS a revised method”

Line 60: should be “multiple sperm” because SPERM is plural

Table 1 last column lower: should be: “Signal analysis standards and accuracy vary between laboratories

Line 79: replace “normal blood cells” with “normal SOMATIC cells”

Line 80: should be “difficult for probes to hybridize to sperm chromatin”

Line 81: should be: “necessary to DECONDENSE sperm nuclei..”

 Response to Comment 4:

Thank you for this helpful and accurate comment. We have made revisions according to your suggestions.

Comment 5: 2.4 lines 87-95 does not read good. The two first sentences are redundant. Also, the authors failed to explain the reason why most researches have focused on aneuploidies of human chromosomes 13, 18, 21 and the sex chromosomes. It is not correct that these chromosomes have higher frequency of aneuploidies. It is just that trisomies for these three autosomes and aneuploidies of sex chromosomes are more viable than aneuploidies of other autosomes. Please revise the entire paragraph.

Response to Comment 5:

Thank you for this helpful feedback. Accordingly, we have revised the manuscript as follows.

Previous studies [10] have primarily focused on chromosomes 13, 18, 21, X, Y, and a few autosomes. Notably, many investigators have performed analyses using probes specific for the 13, 18, 21, X and Y chromosomes because these trisomies for these three autosomes and aneuploidies of sex chromosomes are more viable than aneuploidies of other autosomes.

Comment 6: Line 112: should be “the number of sperm with disomy for chromosomes 13,…”

2.8 should be “Males with Robertsonian translocations”

Line 180: should be “UNBALANCED”

 Response to Comment 6:

Thank you for these helpful comments. We have made corrections based on your suggestions.

Comment 7: ‘ Oocytes

Line 279: replace “of the first premeiotic division” with “of meiosis prophase”

Line 285: not clear what is the meaning of the sentence “The oocytes in the metaphase of meiosis II are generally called oocytes.” Something is missing, “secondary oocyte”?

Line 295: replace “two ends” with “two poles”

Line 299: sister chromatids are separated in anaphase, not metaphase

Response to Comment 7:

According to the reviewer’s suggestion, we have made corrections.

Comment 8:

Lines303-304: the sentence “Aneuploidy….” does not make sense. Disjunction is the normal process in meiosis I  should not cause any aneuploidies. Aneuploidies are caused by nondisjunction. Please reword.

Response to Comment 8:

Thank you for this helpful feedback. Incorporating your suggestions, we have made the following revisions:

Aneuploidy in meiosis I is thought to be caused by chromosome non-disjunction, early chromosome non-disjunction(predivision) [41], and late delay

Comment 9:

Lines 111-112: should be “occur”

Lines 320-322: change the Tense in this paragraph from Past to Present.

Line 340: replace “implemented” with “improved”

Line 344: change to “because a limited number of different fluorescent labels can be used simultaneously”

Line 389: change as “Gabriel et al. CONDUCTED polar body analysis…”

Response to Comment 9:

According to the reviewer’s suggestion, we have made relevant corrections.

Comment 10: Embryo

Line 421: remove either “mainly” or “primarily”, redundant

Line 459: the sentence “It is uncommon…” reads odd. Change to “Aneuploidies are uncommon in live births, except for trisomies 21, 18, and 13, and sex chromosome aneuploidies, which are often viable”

Response to Comment 10:

According to the reviewer’s suggestion, we have revised the manuscript.

Comment 11: CONCLUSION

Line 468: should be “SPERM” which is plural.

Line 471: should be “HAMSTER” not “hamsters”

Line 475: remove “in semen parameters”

Lines 475-476: the sentence “Men with chromosomal aberrations have a higher rate of chromosomal aberrations” does not make any sense.

Line 477: should be “…most of which are ANEUPLOIDIES”

Response to Comment 11:

According to the reviewer’s suggestion, we have made relevant corrections.

Comment 12:  Finally, the authors have explained why elder women have higher rate of aneuploidies. However, there is no explanation why younger women also have higher rate of aneuploidies?

 Response to Comment 12:Thank you for this pertinent request. To address this issue, we included the following sentences in the revised text:

Oocytes from young women under 20 years of age tend to present higher rates of chromosome segregation errors than those from women in their 20s and early 30s [2]. In the first years of ovulation, women under 20 years of age experience high rates of chromosome segregation errors where entire homologous chromosomes mis-segregate during meiosis I. However, the mechanism underlying why aneuploidy is more common in the oocytes of young women is unclear, thus warranting further study. (Line 587–601)

Authors’ Concluding Statement:

Dear Reviewers,

We thank you again for your time, consideration, and thoughtful feedback. We look forward to hearing from you and would be happy to make further changes if required.